# The Role of Hypoxia-Inducible Factor-1 Alpha in Renal Disease

**DOI:** 10.3390/molecules27217318

**Published:** 2022-10-28

**Authors:** Huixia Liu, Yujuan Li, Jing Xiong

**Affiliations:** Department of Nephrology, Union Hospital, Tongji Medical College, Huazhong University of Science and Technology, Wuhan 430022, China

**Keywords:** complications, diabetic nephropathy, hypoxia-inducible factor-1α, renal cancer, renal ischemia/reperfusion injury

## Abstract

Partial pressure of oxygen (pO_2_) in the kidney is maintained at a relatively stable level by a unique and complex functional interplay between renal blood flow, glomerular filtration rate (GFR), oxygen consumption, and arteriovenous oxygen shunting. The vulnerability of this interaction renders the kidney vulnerable to hypoxic injury, leading to different renal diseases. Hypoxia has long been recognized as an important factor in the pathogenesis of acute kidney injury (AKI), especially renal ischemia/reperfusion injury. Accumulating evidence suggests that hypoxia also plays an important role in the pathogenesis and progression of chronic kidney disease (CKD) and CKD-related complications, such as anemia, cardiovascular events, and sarcopenia. In addition, renal cancer is linked to the deregulation of hypoxia pathways. Renal cancer utilizes various molecular pathways to respond and adapt to changes in renal oxygenation. Particularly, hypoxia-inducible factor (HIF) (including HIF-1, 2, 3) has been shown to be activated in renal disease and plays a major role in the protective response to hypoxia. HIF-1 is a heterodimer that is composed of an oxygen-regulated HIF-1α subunit and a constitutively expressed HIF-1β subunit. In renal diseases, the critical characteristic of HIF-1α is protective, but it also has a negative effect, such as in sarcopenia. This review summarizes the mechanisms of HIF-1α regulation in renal disease.

## 1. Introduction

Almost 20% of the blood volume circulates through the human kidney, which is the best perfused organ in terms of weight [1]. However, measured oxygen (O_2_) tension is surprisingly low, ranging from 5 mmHg in the medulla to 50 mmHg in the cortex [2]. This difference lies in the unique structure of the renal vasculature. In the kidney, arterial and venous vessels run in close parallel. The oxygen shunt between arterial and venous vessels can bypass the blood circulation, rendering the oxygen tension relatively low in renal tissue and about 10 mmHg in the renal medulla [3]. The oxygen tension in the renal cortex is highly variable, and the average partial pressure of oxygen is approximately 30 mmHg, which decreases significantly with the change in renal perfusion. In fact, the kidney is intrinsically susceptible to hypoxia. This is because the kidney performs its complex transport functions within a relatively narrow range of partial pressure of oxygen (pO_2_), which is very low in the medulla and highly sensitive to hypoxic injury. Renal cells have utilized various molecular pathways to respond and adapt to decreases in renal oxygenation.

In particular, the transcription factor hypoxia-inducible factor (HIF) (including HIF-1, 2, 3) plays a central role in the cellular adaptation to hypoxia, orchestrating a metabolic switch that allows cells to survive in this environment [4]. The activation of HIF-1 transcription factor is the most recognized pathway adopted by hypoxic cells [5,6,7]. HIF-1 is a heterodimer that is composed of an O_2_-regulated HIF-1α subunit and a constitutively expressed HIF-1β subunit. HIF-1α is rapidly degraded under normoxic conditions, when it is hydroxylated at two proline residues within the oxygen-dependent degradation domain by prolyl hydroxylase domain (PHD) proteins 1–3 [8]. It is subsequently ubiquitinated by the E3 ubiquitin ligase activity of the von Hippel-Lindau tumor suppressor protein (pVHL), thereby targeting HIF-1α for degradation by the proteasome. Hypoxia conditions inhibit prolyl hydroxylation, leading to the stabilization of HIF-1α [9]. HIF-1α then transfers to the nucleus and heterodimerizes with HIF-1β. The HIF-1 dimer subsequently binds to the hypoxia-response element sites on the DNA to initiate the expression of more than 100 genes involved in hypoxia adaptation (Figure 1).

The combination of limited tissue oxygen supply and high oxygen demand is considered a major contributor to the vulnerability of the kidney to acute ischemic injury [10,11]. Hypoxia has long been recognized as an important factor in the pathogenesis of acute kidney injury (AKI), especially renal ischemia/reperfusion injury (RIRI). Convincing evidence has demonstrated that the increased HIF-1 is a landmark change in RIRI [12]. It is generally believed that HIF-1α partially plays a protective role in RIRI by increasing the expression of HIF-1α-target genes involved in the transition from glucose metabolism to glycolysis, scavenging reactive oxygen species (ROS), and regulating cell survive. However, the mechanisms of HIF-1α in RIRI are diverse [13,14,15]. In recent years, accumulating evidence suggests that hypoxia also plays an important role in the pathogenesis and progression of chronic kidney disease (CKD) and CKD-related complications, such as anemia, cardiovascular events, and sarcopenia [16]. In chronic hypoxia, glycolysis not only becomes the main form of energy production, but hypoxia also changes gene expression patterns. In general, the stabilization of HIF and transcriptional activation of hypoxia-induced genes are central mechanisms for adaptation to chronic hypoxia [17]. Animal models have been extensively used to clarify the pathogenesis and underlying mechanisms of renal disease. CKD models mainly include diabetic nephropathy (DN) and are the leading cause of end-stage renal disease (ESRD). It has been reported that HIF-1α activation in tubular cells plays an important role against kidney injury in DN, but the exact mechanisms are incompletely understood [18].

CKD-related complications include anemia, cardiovascular events, and sarcopenia, which are all closely associated with hypoxia. Insufficient production of erythropoietin (EPO) is one of the most important pathological mechanisms leading to anemia in CKD patients. EPO production is mainly stimulated by hypoxia. It has been demonstrated that targeting HIF is effective and well tolerated for the correction of anemia with CKD [19]. CVD is the leading cause of death in patients with CKD. Hypoxia is also a contributing factor to myocardial infarction, cardiac remodeling, atherosclerosis, and peripheral arterial disease in CKD patients. Under hypoxia, HIF signaling affects the development, metabolic response, ischemia and atherosclerosis of heart disease in myriad ways [20]. In addition, HIF-1α mainly plays a major protective role in heart disease [21]. Sarcopenia, a physiological reduction of muscle mass and strength, is one of the complications in patients with CKD. Skeletal muscle hypoxia is thought to be responsible for muscle atrophy and reduced contractility. Mechanistically, HIF-1α contributes to sarcopenia through the glucagon-like peptide-1 (GLP-1) and nuclear factor kappa-B (NF-kB) catabolic pathway [22,23]. Renal cancer is the third most common urologic cancer. It was found that hypoxia is a common character in many types of solid tumors and HIF-1 has been recognized as an important cancer drug target [24]. Mutational inactivation of VHL is the earliest genetic event in renal cancer, leading to the accumulation of HIF-1α and HIF-2α transcription factors. Researches have determined HIF-1α to be an inhibitor and HIF-2α a promoter of aggressive tumor behaviors [25].

Prolyl hydroxylase (PH) enzymes control the degradation of HIF and HIF-PH inhibitors (HIF-PHIs) can correct anemia in patients with renal disease and in animal models of anemia and kidney disease. In addition, HIF-PHIs may provide renal and cardiac protection to patients suffering from kidney disease with metabolic syndrome [26].

This review provides a detailed summary of the regulatory mechanisms of HIF-1α in the kidney diseases described above.

## 2. Different Pathophysiological Mechanisms of HIF-1α in Renal Disease

### 2.1. The Role of HIF-1α in RIRI

Currently, IR is the most widely used model for clinical AKI and renal transplant studies. Bioinformatic analysis of the GEO dataset and integration of gene expression profiles in a rat model of renal IRI identified HIF-1α signaling [15]. Mitochondrial dysfunction, induction of inflammation, apoptosis, autophagy, necroptosis and oxidative stress are the major factors in the pathogenesis of RIRI, and HIF-1α can protect against the kidney injury caused by RIRI through the above mechanisms (Figure 2).

B-cell lymphoma-2 (BCL-2) interacting protein 3 (BNIP3) is a mitochondrial outer membrane protein that is closely related to mitophagy, and it is also a target gene of HIF-1, which can be transcriptionally activated by HIF-1 in hypoxia. Fu et al. conducted in vivo and in vitro studies and demonstrated that HIF-1α-BNIP3-mediated mitophagy in tubular cells plays a protective role through the inhibition of apoptosis and oxidative stress in RIRI [27]. In addition, the intrinsic mechanisms implicated in RIRI-associated cell death include the disruption of mitochondrial dynamics, resulting in mitochondrial fragmentation. Wei et al. revealed that microRNA-668 (miR-668), previously found up-regulated in the RIRI model, has a protective effect during RIRI by inhibiting pathogenic mitochondrial fragmentation and the subsequent apoptosis of renal tubular cells [28]. Moreover, the induction of miR-668 in RIRI is HIF-1 dependent, while among the miR-688 target genes, only mitochondrial protein 18kDa (MTP18) is associated with mitochondrial dynamics. Therefore, the final results indicated that HIF-1α plays a protective role in RIRI through the HIF-1α/miR668/MTP18 axis by inhibiting pathogenic mitochondrial fragmentation and cell apoptosis [29].

RIRI is accompanied by a strong inflammatory response. Li et al. found that renal tubular HIF-1α expression was significantly increased at the transcriptional level in the RIRI model, which is closely associated with macrophage-dependent inflammation. The nuclear factor kB (NF-kB) plays a central role in the inflammatory response. Mechanistically, the inflammatory cytokine-mediated NF-kB pathway activation can directly bind to the HIF-1α promoter and enhance its transcription, thus protecting against RIRI [30]. Dexmedetomidine (Dex) is a kind of clinically common sedative, and it has been determined that it promotes the recovery of renal function and reduces the inflammatory level in RIRI rats through the phosphatidylinositol 3-hydroxy kinase (PI3K)/protein kinase B(Akt)/HIF-1α signaling pathway [14].

Apoptosis, autophagy and necroptosis are the important pathological mechanisms of RIRI. The NOD-like receptor thermal protein domain-associated protein 3(NLRP3) was reported to be involved in RIRI and NLRP3 knockout could protect against RIRI. Tetramethylpyrazine (TMP), a calcium antagonist with various pharmacological effects, has been widely used in the treatment of AKI. Sun et al. demonstrated that TMP can reduce NLRP3 protein expression in renal tissue and tubular cell apoptosis by reducing HIF-1α expression and improving renal function to alleviate RIRI [31]. Autophagy is activated by different types of stress, such as ischemia and inflammation, and is involved in AKI. Lipocalin 2 (Lcn2) is an important marker of renal injury and its production markedly increases in response to stimulation such as ischemia [32]. Mechanistically, Qiu et al. found that recombinant Lcn2 attenuated hypoxia-induced apoptosis and the downregulation of HIF-1α blunted Lcn2-induced autophagy and enhanced apoptosis. What’s more, Lcn2 attenuated NF-kB subunit p65 activation under hypoxia conditions. Thus, Lcn2 could protect against RIRI in mice through autophagy activation mediated by HIF-1α and NF-kB crosstalk [31]. Necroptosis is also a major contributor to the pathogenesis of ischemic AKI. Necrostatin-1 (Nec-1), an inhibitor of the kinase domain of receptor-interacting protein kinase-1 (RIP1), was previously reported to protect against RIRI. In previous studies, miR-26a has been identified as a potential novel target for the treatment RIRI. Moreover, previous bioinformatic studies have found that the transient receptor potential cation channel (TRPC6) is an upregulated and differentially expressed gene involved in the pathogenesis of I/R injury, and showed that it can prevent RIRI by inhibiting necroptosis. Shen et al. suggested that Nec-1 can effectively protect against RIRI by inhibiting necroptosis, oxidative stress, and inflammation, and may exert its effects through the mediation of the HIF-1α/miR-26a/TRPC6/PARP1 signaling pathway in vivo and vitro [33].

In addition, angiogenesis is one of the compensatory responses after RIRI. Promoting angiogenesis after reperfusion can prevent RIRI and improve the prognosis of AKI. HIF plays an important role in angiogenesis by upregulating vascular endothelial growth factor (VEGF) [34]. It has been reported that miR-21, induced by HIF, is essential in the process of vascular remodeling or angiogenesis in endothelial cells [35]. Thrombospondin 1 (TSP-1), is a multifunctional protein and was initially recognized as an endogenous inhibitor of angiogenesis. Xu et al. concluded that HIF-1α induced angiogenesis by upregulating not only vascular endothelial growth factor but also miR-21 via inhibiting a novel target gene TSP-1 contributing to the protective effect of HIF-1α on RIRI [36].

### 2.2. The Role of HIF-1α in Diabetic Nephropathy

DN is one of the most common models of CKD and the leading cause of ESRD. Earlier studies have demonstrated that hypoxia is an early event in the development and progression of experimental DN, and an increased HIF-1α expression in diabetic kidneys compared to the kidneys of control rats and normal human kidneys [37,38,39] (Figure 3). Ferroptosis is a recently discovered form of iron-dependent cell death. Heme is a main source of synthetic iron, and heme oxygenase (HO)-1 metabolizes heme into biliverdin/bilirubin, carbon monoxide, and ferrous iron. HO-1 can be induced by a variety of cues, including inflammatory mediators, oxidants, and physical or chemical stimuli, and HO-1 is one of the HIF target genes [40]. Chronic hypoxia due to renal ischemia induces the increase of HIF-1α in renal tubules of diabetic models, with the elevated HO-1 level [41,42]. Feng et al. indicated that ferroptosis might enhance DN and damage renal tubules in DN models through the HIF-1α/HO-1 pathway [43]. However, a study in 2020 demonstrated that HIF-1α improved mitochondrial dysfunction and restricted mitochondria-dependent apoptosis in the tubular cells of DN via the HO-1 pathway. In addition, the HIF-1α/HO-1 pathway is the pivotal pathway mediating tubular cell mitochondrial dynamics in DN [18].

Noncoding RNAs, arbitrarily separated into small ncRNAs and long noncoding RNAs (lncRNAs), are involved in multifarious physiological and pathological processes. A growing number of studies suggest that most genes are regulated by small ncRNAs, such as miRNA, and that miRNAs are involved in the pathogenesis of tubular injury in DN. The current study has demonstrated that miR-122-5p is up-regulated in the renal tubular cells of DN, and that it alleviates renal tubular cell death and kidney injury in DN. Factor inhibiting hypoxia-inducible factor-1 (FIH-1) was a direct target of miR-122-5p. Therefore, it indicated that miR-122-5p could ameliorate tubular injury in diabetic nephropathy via the FIH-1/HIF-1α pathway [44]. Numerous lncRNAs are reported to be involved in various kinds of diseases and oncogenesis, including DN [45]. By using RNA sequencing, we found many lncRNAs expressing differently in DN models, including the decreased expression level of lincRNAs-1700020I14Rik in DN in vitro and in vivo. In addition, increasing evidence shows that silent information regulator T1 (Sirt1) is involved in many important biological processes, including inflammation, renal interstitial fibrosis, autophagy under hypoxia condition, ageing, and oxidative stress. A decreased expression of Sirt1 was found in DN, and knockdown of Sirt1 expression may abolish the beneficial effects of the active component against renal damage in DN. Silencing Sirt1 could promote fibrosis factors and inflammation factors in DN glomerular measangial cells through promoting its downstream molecule HIF-1α expression. The results suggested that 1700020I14Rik plays an important regulatory role in DN through miR-34a-5p/Sirt1/HIF-1α signaling [46].

Hypoxia-induced inflammation plays a central role in the DN. Studies have shown that macrophage polarization occurs in renal inflammation and plays a decisive part in DN. Transforming growth factor β-activated kinase 1-binding protein 1 (TAB1) is a specific protein that interacts with transforming growth factor β-activated kinase 1 (TAK1). TAB1/TAK1 can activate NF-kB in bone marrow-derived macrophages (BMMs) of DN, thus upregulating HIF-1α activity and enhancing glycolytic metabolism. Zeng et al. concluded that the TAB1/NF-kB/HIF-1α signaling pathway regulates glycolysis and activation of macrophages in DN [47]. Previous studies have shown that high glucose-mediated tubulointerstitial accumulation of extracellular matrix (ECM) has an important role in the pathogenesis of DN. The HIF-1α/VEGF signaling pathway has previously been shown to be involved in the regulation of the ECM. Hirudin is an anticoagulant produced by the salivary glands of the medicinal leech. Hirudin reduces the deposition of ECM in DN models through the inhibition of the HIF-1α/VEGF pathway to protect kidney function and delay disease progression [48].

### 2.3. The Role of HIF-1α in Chronic Kidney Disease-Related Complications

#### 2.3.1. Renal Anemia

Anemia is a common complication of CKD, mainly due to injured kidneys failing to produce sufficient amounts of EPO, which regulates red blood cell production. Hypoxia serves as the major stimulus of EPO production. Iron is essential in erythropoiesis. Iron deficiency also occurs in CKD patients due to the inadequate provision or absorption of dietary iron (Figure 4). HIF-1α is mainly induced in renal tubular and glomerular epithelial cells and in papillary interstitial cells, whereas HIF-2α is expressed in endothelial cells and fibroblasts upon hypoxic stimulation, suggesting that HIF-2α is the main regulator of EPO production [49]. Under hypoxic conditions, HIF-2α regulates EPO expression in combination with hypoxia response elements on the EPO gene in the kidney and liver [50]. Voit et al. revealed that EPO production in the kidney can also be regulated by HIF-1α, which is degraded under normoxic conditions by HIF-prolyl hydroxylase (HIF-PHD) [19]. The discovery of prolyl hydroxylase domain (PHD) enzymes as regulators of hypoxia-inducible factor (HIF)-dependent erythropoiesis has led to the development of novel therapeutic agents for renal anemia [51]. Roxadustat, the first small-molecule PHI, can lead to increased EPO production, better iron absorption, and amelioration of anemia in CKD [19]. In addition, HIF-2α also regulates iron metabolism by stimulating duodenal cytochrome B (DCYTB) and divalent metal transporter-1 (DMT1) expression.

#### 2.3.2. Cardiovascular Disease

CVD is the main cause of death in patients with CKD. Hypoxia is also a promoting factor of myocardial infarction, cardiac remodeling, atherosclerosis and peripheral arterial disease in CKD patients. Under hypoxia, HIF signaling affects the development, metabolic response, ischemia and atherosclerosis in heart disease in myriad ways (Figure 4). Animal experiments have shown that the HIF-1α deletion has a wide range of cardiac abnormalities, including septation and trabeculation, and is affected by specific alleles and the mouse genetic background [52]. Thus, HIF-regulated transcription is necessary for important cardiac developmental events. Cellular metabolism is inextricably linked with cardiac contractility and HIF plays a crucial role in shaping the metabolic response in the heart. HIF-1α promotes glycolysis during hypoxia and influences mitochondrial metabolism to proper cardiometabolic function [20]. As a vasodilator, NO plays a vital role in the regulation of vascular tone through the regulating cGMP in smooth muscle cells, S-nitrosylation of target proteins, the activation of sarco/endoplasmic reticulum calcium ATPase and the production of cyclic inosine monophosphate. HIF-1α can activate inducible nitric oxide synthase (iNOS) gene expression by increasing NO synthesis to limit ischemic damage in the heart [53]. The effects of HIF-1α in the three most important cell types of atherosclerosis--macrophages, vascular smooth muscle cells (VSMCs), and endothelial cells (ECs) are essential. Whereas HIF-1α directly induces VEGF, endothelin-1, and matrix metalloproteinases (MMPs) in endothelial cells to facilitate angiogenesis, its effect on vascular smooth muscle cells is to induce proliferation of these cells in the atheroma by up-regulating factors such as CD98 and macrophage migration inhibitory factor (MIF). HIF-1α also modulates the function of diseased macrophage foam cells by making the cells more inflammatory and apoptotic while simultaneously inhibiting their ability to metabolize lipids [20].

#### 2.3.3. Sarcopenia

Sarcopenia is one of the complications of CKD patients. Hypoxia induces oxygen delivery reduction to the muscles, which is more obvious in anemia with CKD. Mechanistically, HIF-1α deletion stimulates GLP-1 secretion in human adipocytes, which contributes to sarcopenia in hypoxia [54]. Furthermore, hypoxia induces muscle wasting by activating the HIF-1α and NF-kB catabolic pathways and inhibiting the anabolic mammalian target of the rapamycin (mTOR) pathway [55,56]. Cirillo et al. revealed a decrease in the expression of HIF-1α and its target genes in sarcopenia, as well as of PAX7, the major stem cell marker of satellite cells, whereas the atrophy marker MURF1 was increased. In addition, a pharmacological activation of HIF-1α and its target genes caused a reduction in skeletal muscle atrophy and activation of PAX7 gene expression. Thus, HIF-1α plays a role in sarcopenia and is involved in satellite cell homeostasis but need further research [57] (Figure 4).

### 2.4. The Role of HIF-1α in Renal Cancer

Hypoxia is common in many types of solid tumors, in which tumor cells rapidly proliferate and form large solid tumor masses, resulting in the obstruction and compression of the blood vessels around these masses. These abnormal vessels usually do not function properly, causing poor O_2_ supply to the central tumor region [58]. Renal cancer is a common urological malignancy with limited therapeutic options for metastatic disease. Most clear cell renal cell carcinomas (ccRCC) are associated with loss of von Hippel-Lindau tumor suppressor (pVHL) function and deregulation of the hypoxia pathway. The VHL tumor suppressor is inactivated in a majority of ccRCC tumors. VHL is the substrate recognition component of an E3 ubiquitin ligase complex containing the elongins B and C, Cullin-2, and Rbx1 that targets the hydroxylation, oxygen-sensitive *α*-subunits of HIFs for ubiquitination and degradation by the 26S proteasome. The VHL loss results in the constitutive activation of HIF targets, including the pro-angiogenic factors VEGF and platelet derived growth factor (PDGF) [59]. Activated HIF-1 plays a crucial role in the tumor cell adaptive response to oxygen changes by transcriptionally activating more than 100 downstream genes that regulate important biological processes required for tumor survival and progression. Further studies on the HIF would help in developing cancer therapies [60,61] (Figure 5).

#### 2.4.1. The Tumor-Suppressor Role of HIF-1α in Renal Cancer

The HIF-1α is present on chromosome 14q, which is frequently deleted in the ccRCC. Many ccRCC either do not have or produce very low levels of HIF-1α, and tumors with a 14q deletion show transcriptional features, indicating reduced HIF-1α activity. Functional evidence that hypoxia-inducible transcription factors are important for clear cell renal cancer HIF-1α and HIF-2α subunits display distinct but overlapping target gene repertoires [62,63]. Genomic analyses of renal tumors have revealed that deletions in a region of chromosome 14 that harbors the HIF-1α gene are a common feature of ccRCC [64]. In line with this finding, several commonly used ccRCC cell lines (e.g., 786-O and A498) have lost the expression of the full-length HIF-1α. This re-expression of HIF-1α also leads to the transcriptional activation of the established HIF-1α target genes. This suggests that in the context of ccRCC, HIF-1α plays a tumor-suppressor role, while genetic and functional data suggest that HIF-2α and HIF-1α play opposite roles in ccRCC biology. However, Hoefflin, R. et al. revealed that HIF-1α is required for tumor formation, whereas HIF-2α deficiency has only a moderate effect on tumor genesis and growth in the mouse ccRCC model. It is suggested that HIF-1α and HIF-2α are likely to play different roles at different stages of tumour formation and progression and that these roles might potentially be modulated by the spectrum of mutations present in each individual ccRCC [25].

#### 2.4.2. The HIF-1α/CPT1A Pathway and HIF-1α/HO-1 Pathway in Renal Cancer

The principal roles of fatty acid (FA) are to serve as substrates for membrane synthesis, energy stores, and the production of signaling molecules. Abnormal cancer metabolism leads to changes in decisions regarding FA fates, including the altered balance in ccRCC toward excessive storage in the form of lipids. Carnitine palmitoyltransferase 1A (CPT1A) is the rate-limiting enzyme of the FA transport system controlling entry into the mitochondrion and is a hypoxia-repressed target gene that regulates lipid accumulation in ccRCC. Elevated CPT1A expression limits ccRCC tumors growth, while CPT1A is inhibited by HIF1 and HIF2, reducing FA transport into the mitochondria, thus implying that HIF controlling FA metabolism is required for ccRCC tumorigenesis [65]. In addition, glutamine is essential for many fundamental functions in cancer cells, such as producing antioxidants to remove ROS, maintaining mitochondrial metabolism, and activating cell signaling [66]. Mitochondrial protein sirtuin 4 (SIRT4) is critical for regulating the mitochondrial glutamine metabolism. HO-1 is a cytoprotective molecule with antioxidant, anti-inflammatory, and anti-apoptotic properties, and targeting HO-1 has been implemented as an anti-tumor therapy in several studies [67]. It has been demonstrated that SIRT4 was downregulated in ccRCC. Results from the TCGA database indicate that HO-1 mRNA expression negatively correlated with SIRT4. Ying et al. revealed that VHL regulates the sensitivity of ccRCC to SIRT4-mediated metabolic stress via the HIF-1α/HO-1 pathway [68].

#### 2.4.3. The ENTPD3-AS1/miR-155-5p/HIF-1α Axis in Renal Cancer

In the past decade, accumulating evidence suggests that several lncRNAs play a crucial role in the pathogenesis of ccRCC. These lncRNAs participate in the biological process of RCC through HIF-related and HIF-independent pathways. lncRNA ENTPD3-AS1, a long non-coding RNA located at 3p, was found in studies with low expression detected in ccRCC tissues and associated with poor prognosis, suggesting its tumor-suppressive role in ccRCC [69]. Moreover, ENTPD3-AS1 can directly interact with miR-155-5p, leading to the upregulation of HIF-1α, thus inhibiting the occurrence of ccRCC. Furthermore, thousands of disease-related SNPs have been identified by GWAS in the last decades, most of which are located in noncoding regions. Previous meta-analysis indicated that SNP rs67311347 at 3p22.1 was confirmed to be associated with the risk of ccRCC [70]. Wang et al. showed that the G > A mutation of rs67311347 created a binding motif of ZNF8 and subsequently upregulated ENTPD3-AS1 expression by acting as an enhancer. Thus, the protective SNP rs67311347 suppresses the occurrence of RCC through an ENTPD3-AS1/miR-155-5p/HIF-1α axis [71].

## 3. Conclusions

Hypoxia is a critical factor in the pathogenesis of kidney diseases, including AKI, CKD, CKD-related complications, and renal cancer. The transcription factor HIF-1 is a master regulator of adaptive responses to hypoxia. HIF-1 is composed of the HIF-1α subunit and HIF-1β subunit. In contrast to HIF-β subunits, the expression level of HIF-1α is tightly regulated by changes in oxygen concentration through proteolytic degradation and transcriptional regulation. HIF-1α controls oxygen delivery by regulating angiogenesis and vascular remodeling, and oxygen utilization by regulating glucose metabolism and redox homeostasis. Currently, many analyses of animal models suggest the different mechanisms of HIF-1α regulation in the RIRI, DN, and ccRCC models. In these models, the critical characteristic of HIF-1α is protective, but it also has a negative effect, such as in sarcopenia. This review summarizes the mechanisms of HIF-1α regulation in animal models of renal disease, which not only provides new insight with regard to understanding the pathophysiology of kidney disease, but also provides rational strategies for therapeutic intervention.

## Figures and Tables

**Figure 1 molecules-27-07318-f001:**
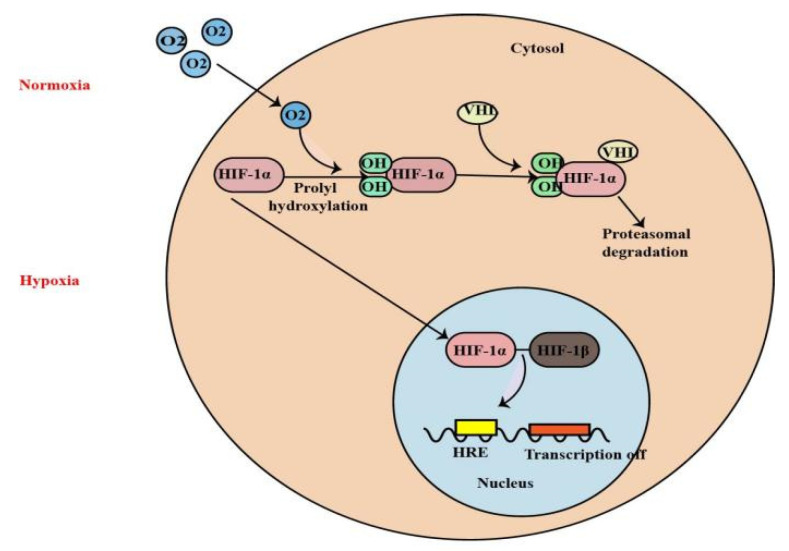
Under normoxia, HIF-1α is rapidly degraded when it is hydroxylated at two proline residues within the oxygen-dependent degradation domain by prolyl hydroxylase domain (PHD) proteins. It is subsequently ubiquitinated by the E3 ubiquitin ligase activity of the von Hippel-Lindau tumor suppressor protein (pVHL), thereby targeting HIF-1α for degradation by the proteasome. Under hypoxia, HIF-1α transfers to the nucleus and heterodimerizes with HIF-1β. The HIF-1 dimer subsequently binds to the hypoxia-response element sites on the DNA to initiate the expression of more than 100 genes involved in hypoxia adaptation.

**Figure 2 molecules-27-07318-f002:**
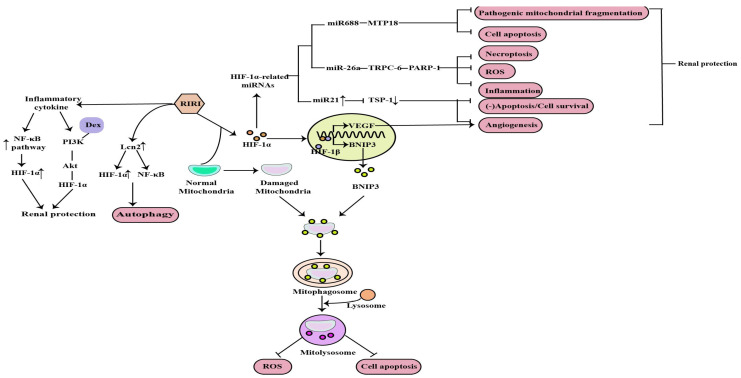
Mitochondrial dysfunction, induction of inflammation, apoptosis, autophagy, necroptosis, angiogenesis and ROS are the major factors in the pathogenesis of RIRI, and HIF−1α can protect against the kidney injury caused by RIRI through the above mechanisms (↑: elevated expression; ↓: decreased expression).

**Figure 3 molecules-27-07318-f003:**
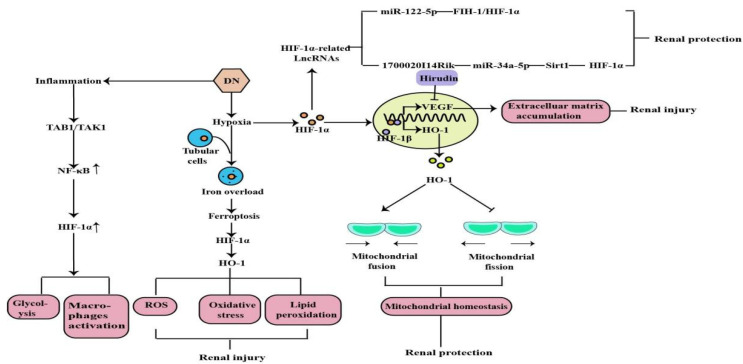
Mitochondrial dysfunction, lipid peroxidation, oxidative stress, ROS, machrophages activation, glycolysis, and the induction of inflammation are the major factors in the pathogenesis of DN, and HIF-1α can protect and aggravate the kidney injury caused by DN through the above mechanisms(↑: elevated expression).

**Figure 4 molecules-27-07318-f004:**
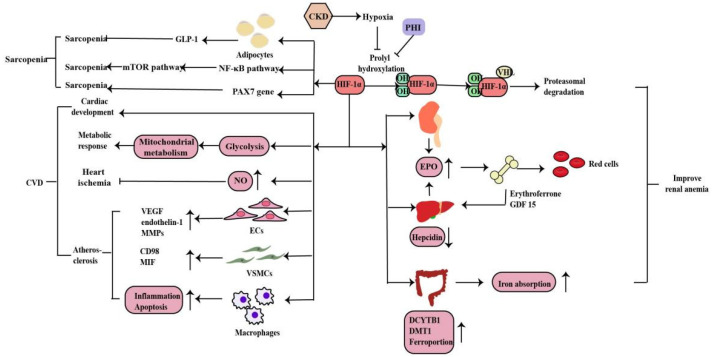
HIF-1α regulates EPO gene in the kidney and liver and also regulates iron metabolism by stimulating DCYTB and DMT1 expression. Inhibition of HIF-PHD leads to increased EPO production, better iron absorption, and amelioration of anemia in CKD. Under hypoxia, HIF signaling affects the cardiac development, metabolic response, heart ischemia and atherosclerosis to CVD in myriad ways. And HIF-1α plays a dual role in sarcopenia through the mechanisms described above (↑: elevated expression; ↓: decreased expression).

**Figure 5 molecules-27-07318-f005:**
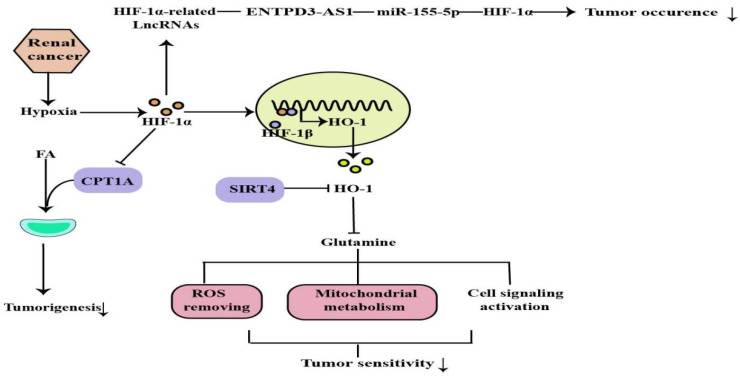
ROS removing, mitochondrial metabolism, cell signaling activation are regulated by HIF-1α, which can regulate tumor occurrence and sensitivity through the above mechanisms (↓: decreased expression).

## Data Availability

Not applicable.

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
