# Peer review of "The Role of Hypoxia-Inducible Factor-1 Alpha in Renal Disease"

_molecules, 2022, doi:10.3390/molecules27217318_

Round 1
Reviewer 1 Report
1. Please cite more articles from the year 2020-2022. Many articles publish recently discussed HIF stabilization in renal diseases.
2. 2.4.1 and 2.4.2 present the same discussion of the tumor-suppressor role of HIF-1α in renal cancer.
3. 2.3.1 Renal anemia : please add more discussion of HIF1α roles and the newest treatment and evidence for anemia, such as HIF-prolyl hydroxylase inhibitors (PHI)
4. 2.3.2 Cardiovascular disease: Please provide the newest evidence of the regulation of HIF1a in Cardiovascular diseases.
5. 2.3.3 Sarcopenia: the role of HIF1α still lacks the newest evidence. Please read this article: Human Sarcopenic Myoblasts Can Be Rescued by Pharmacological Reactivation of HIF-1, International journal of molecular sciences (2022).
6. As mentioned in the title “The role of hypoxia-inducible factor-1 alpha in renal disease”, please introduce the critical characteristic of HIF-1α in the regulation of renal disease in the abstract.
7. Abstract: pO2; please explain this abbreviation, what is “p” means here, phosphorylation or others.
8. Please explain the GFR abbreviation in the abstract.
9. Please explain the importance of HIF-1α roles in renal disease with the illustration (figures) based on this review article analysis
10. Please revise the conclusion:
a) Line 334: “The transcription factor HIF-1 is a master regulator of adaptive responses to hypoxia in all metazoan species,...” the authors did not discuss the renal disease in various species and no explanation about metazoans in this manuscript. Thus, this sentence cannot be accepted as a conclusion.
b) Line 336-340: the critical role of HIF-1α in kidney disease conclusion is prevalent and general. Please conclude the author reviews explicitly on the new light for understanding the pathophysiology of renal disease.
Author Response
Response to Reviewer
We appreciate the reviewer’ thoughtful reviews and helpful comments which helped us improve this manuscript significantly. In the revision, we have addressed all the concerns according to your suggestions. The changes in the revised manuscript were marked up using the “Track Changes” function.
Reviewer 1:
- Please cite more articles from the year 2020-2022. Many articles publish recently discussed HIF stabilization in renal diseases.
ANSWER: Thank you for your helpful advice. We have cited more articles from the year 2020-2022 in the revised manuscript.
- 2.4.1 and 2.4.2 present the same discussion of the tumor-suppressor role of HIF-1α in renal cancer.
ANSWER: Thank you for the suggestion. We have removed 2.4.2 from the initial version.
- 2.3.1 Renal anemia: please add more discussion of HIF-1α roles and the newest treatment and evidence for anemia, such as HIF-prolylhydroxylase inhibitors (PHI) .
ANSWER: We appreciate your advice. We have added more discussion of HIF-1α roles in renal anemia and therapeutic mechanism of PHIs in 2.3.1.
- 3.2 Cardiovascular disease: Please provide the newest evidence of the regulation of HIF-1αin Cardiovascular diseases.
ANSWER: Thanks for your helpful advice. We have provided the newest evidence of the regulation of HIF-1α in cardiovascular diseases, including development, metabolic response, ischemia and atherosclerosis in 2.3.2.
- 2.3.3 Sarcopenia: the role of HIF1α still lacks the newest evidence. Please read this article: Human Sarcopenic Myoblasts Can Be Rescued by Pharmacological Reactivation of HIF-1, International journal of molecular sciences (2022).
ANSWER: Thanks for your good suggestion. We have revised the contents and cited this reference in 2.3.3.
- As mentioned in the title “The role of hypoxia-inducible factor-1 alpha in renal disease”, please introduce the critical characteristic of HIF-1α in the regulation of renal disease in the abstract.
ANSWER: Thank you for your suggestion. We have introduced the critical characteristic of HIF-1α in the regulation of renal disease in the Abstract.
- Abstract: pO2; please explain this abbreviation, what is “p” means here, phosphorylation or others.
ANSWER: Thank you for your helpful advice. We have explained the implications of pO2 in the Abstract.
- Please explain the GFR abbreviation in the abstract.
ANSWER: Thank you for the suggestion. We have explained the implications of GFR in the Abstract.
- Please explain the importance of HIF-1α roles in renal disease with the illustration (figures) based on this review article analysis.
ANSWER: We appreciate your advice. We have added a figure for a better understanding of HIF-1α roles in renal disease, and that is Figure 1.
- Please revise the conclusion:
- a) Line 334: “The transcription factor HIF-1 is a master regulator of adaptive responses to hypoxia in all metazoan species,...” the authors did not discuss the renal disease in various species and no explanation about metazoans in this manuscript. Thus, this sentence cannot be accepted as a conclusion.
- b) Line 336-340: the critical role of HIF-1α in kidney disease conclusion is prevalent and general. Please conclude the author reviews explicitly on the new light for understanding the pathophysiology of renal disease.
ANSWER: Thanks for your helpful advice. Based on the reviewer's comments, we have revised our conclusions. We have removed the contents about metazoans and concluded this review more explicitly.
Reviewer 2 Report
Journal of molecules
Research Article;
The article entitled “The role of hypoxia-inducible factor-1 alpha in renal disease’’. The authors best explain the role of Hypoxia in Renal diseases. Oxygen maintains in the kidney in several ways. The susceptibility of these dealings reduces the kidney's susceptibility to hypoxic injury which result is the cause of different renal diseases. Hypoxia has long been recognized as an important factor in the pathogenesis of acute kidney injury, especially renal ischemia/reperfusion injury. Amassing evidence recommends that hypoxia also plays a key role in the pathogenesis and progression of chronic kidney disease and related complications, such as anaemia, cardiovascular events, and sarcopenia. In addition, renal cancer is linked to the deregulation of hypoxia pathways. Renal utilizes various molecular pathways to respond and adapt to changes in renal oxygenation. Particularly, hypoxia-inducible factor HIF including HIF-1, 2, and 3 are activated in renal disease and play a major role in the protective response to hypoxia. HIF-1 is a heterodimer that is composed of an oxygen-regulated HIF-1α subunit and a constitutively expressed HIF-1β subunit. This review summarizes the mechanisms of HIF-1α regulation in renal disease
I carefully read the manuscript and found it suitable for publication in the journal. I accept this article for possible publication. There are some minor mistakes in the article which should be corrected by the authors. After the correction of all the mistakes and revision, the article could be considered for publication in the prestigious molecules Journal.
Comments for Authors
Ø At least add a graphical image.
Ø Write the complete word in pO2 (line 8)
Ø Write complete word GFR (line 9)
Ø In the section abstract, Write oxygen instead of O2.
Ø Write keywords in alphabetical order.
Ø Section Introduction; Revise it. The authors want to put more latest related citations in the introduction part. must include the citation after 2015.
Ø Write complete names instead of DN and CKD, in subtitles 2.2, 203 respectively (line 159, 209)
Ø The author needs to remove space from HIF-1 α (HIF-1α) in subtitles. Revised all the titles.
Ø Use EndNote or Mendeley software for reference sequences.
Ø Check grammatically and spelling throughout the manuscript. There are some mistakes.
Ø The needs to clarify more (High levels of HIF are particularly important in the clear cell type of kidney cancer and HIF-1α and HIF-2α both have effects on renal cancer evolution). This sentence is not clear about its meaning. (line 88, 89)
Ø The author needs to Revise the manuscript carefully, as there are many words with missing space.
Ø It is unnecessary to write protein kDa in the review article. line 100 shows the BCL-2 kDa, what does this mean? Also, write the abbreviation of BCL-2
Ø The need to write in vivo and in vitro in italic style.
Ø As the author mentioned the reference there is no need to write the name of the reference author. The author needs to revise it.
Ø The author needs to include a graphical image of the HIF-1α pathway during normoxia and hypoxia.
Ø The needs to include one short paragraph about the HIF-1α inhibitor and its effect on HIF-1α.
Cite the following references;
1, https://doi.org/10.1038/s41419-021-03771-z2, https://doi:10.1016/j.ebiom.2022.103942
3, https://doi.org/10.2174/1871520622666220831124321
Author Response
Response to Reviewer
We appreciate the reviewer’ thoughtful reviews and helpful comments which helped us improve this manuscript significantly. In the revision, we have addressed all the concerns according to your suggestions. The changes in the revised manuscript were marked up using the “Track Changes” function.
Reviewer 2:
I carefully read the manuscript and found it suitable for publication in the journal. I accept this article for possible publication. There are some minor mistakes in the article which should be corrected by the authors. After the correction of all the mistakes and revision, the article could be considered for publication in the prestigious molecules Journal.
- At least add a graphical image.
ANSWER: Thank you for your helpful advice. We have added a figure for a better understanding of HIF-1α roles in renal disease in the revised manuscript.
- Write the complete word in pO2 (line 8).
ANSWER: Thank you for the suggestion. We have written the complete word of pO2.
- Write complete word GFR (line 9).
ANSWER: We appreciate your advice. We have written the complete word of GFR.
- In the section abstract, Write oxygen instead of O2.
ANSWER: Thanks for your helpful advice. We have written oxygen instead of O2 in the Abstract.
- Write keywords in alphabetical order.
ANSWER: Thanks for your good suggestion. We have written keywords in alphabetical order.
- Section Introduction; Revise it. The authors want to put more latest related citations in the introduction part. must include the citation after 2015.
ANSWER: Thank you for your suggestion. We have revised the introduction and cited more articles after the year 2015-2022.
- Write complete names instead of DN and CKD, in subtitles 2.2, 203 respectively (line 159, 209).
ANSWER: We appreciate your advice. We have written complete names of DN and CKD in subtitles 2.2 and 2.3.
- The author needs to remove space from HIF-1 α (HIF-1α) in subtitles. Revised all the titles.
ANSWER: Thanks for your helpful advice. We have revised all the titles to remove space of HIF-1α.
- Use EndNote or Mendeley software for reference sequences.
ANSWER: Thanks for your good suggestion. We have used EndNote software for reference sequences.
- Check grammatically and spelling throughout the manuscript. There are some mistakes.
ANSWER: Thank you for your suggestion. We have checked grammatically and spelling throughout the manuscript.
- The needs to clarify more (High levels of HIF are particularly important in the clear cell type of kidney cancer and HIF-1α and HIF-2α both have effects on renal cancer evolution). This sentence is not clear about its meaning. (line 88, 89)
ANSWER: Thank you for the suggestion. We have clarified more about the roles of HIF-1α and HIF-2α in renal cancer in the revised manuscript. (line302-305)
- The author needs to Revise the manuscript carefully, as there are many words with missing space.
ANSWER: We appreciate your advice. We have added all the missing space in the manuscript.
- It is unnecessary to write protein kDa in the review article. line 100 shows the BCL-2 kDa, what does this mean? Also, write the abbreviation of BCL-2
ANSWER: Thanks for your helpful advice. We have removed the protein kDa in the review article and written complete word of BCL-2.
- The need to write in vivo and in vitro in italic style.
ANSWER: We appreciate your advice. We have written in vivo and in vitro in italic style in the revised manuscript.
- As the author mentioned the reference there is no need to write the name of the reference author. The author needs to revise it.
ANSWER: Thank you for your helpful advice. We revised the format of the references in the revised manuscript.
- The author needs to include a graphical image of the HIF-1α pathway during normoxia and hypoxia.
ANSWER: Thank you for the suggestion. We have added a figure for a better understanding of HIF-1α pathway during normoxia and hypoxia, and that is Figure 1.
- The needs to include one short paragraph about the HIF-1α inhibitor and its effect on HIF-1α.
ANSWER: Thanks for your helpful advice. We have added a short paragraph about the HIF-1α inhibitor and its effect on HIF-1α in the Introduction section.
17.Cite the following references;
1, https://doi.org/10.1038/s41419-021-03771-z
2, https://doi:10.1016/j.ebiom.2022.103942
3, https://doi.org/10.2174/1871520622666220831124321
ANSWER: We appreciate your advice. We have cited the above three references and added the relevant contents.
Round 2
Reviewer 1 Report
Regarding the reviewer's comment (Figure is necessary to be added to the manuscript): The author has added the figure. However, The figure details are general basic of HIF1a roles, prevalent, and did not explain, as mentioned in the abstract, "the negative effect of sarcopenia".
Thus, the detailed description of HIF1a regulation under renal disease conditions cannot be represented by the only figure presented in this manuscript.
Author Response
Response to Reviewer
We appreciate the reviewer’ thoughtful reviews and helpful comments which helped us improve this manuscript significantly. In the revision, we have addressed all the concerns according to your suggestions. The changes in the revised manuscript were marked up using the “Track Changes” function.
Reviewer 1:
Regarding the reviewer's comment (Figure is necessary to be added to the manuscript): The author has added the figure. However, the figure details are general basic of HIF-1α roles, prevalent, and did not explain, as mentioned in the abstract, "the negative effect of sarcopenia". Thus, the detailed description of HIF-1α regulation under renal disease conditions cannot be represented by the only figure presented in this manuscript.
ANSWER: Thank you for your helpful advice. We have added four figures for a better understanding of HIF-1α roles in renal disease, and that is Figure 2, 3, 4 and 5.
Round 3
Reviewer 1 Report
The authors have completed all reviewer comments and suggestions.
The present manuscript is accepted to be published in Molecules journal.